# Exploring the Role of Serotonin as an Immune Modulatory Component in Cardiovascular Diseases

**DOI:** 10.3390/ijms24021549

**Published:** 2023-01-12

**Authors:** Aqeela Imamdin, Emiel P. C. van der Vorst

**Affiliations:** 1Institute for Molecular Cardiovascular Research (IMCAR), RWTH Aachen University, 52074 Aachen, Germany; 2Aachen-Maastricht Institute for CardioRenal Disease (AMICARE), RWTH Aachen University, 52074 Aachen, Germany; 3Interdisciplinary Center for Clinical Research (IZKF), RWTH Aachen University, 52074 Aachen, Germany; 4Institute for Cardiovascular Prevention (IPEK), Ludwig-Maximilians-University Munich (LMU), 80336 Munich, Germany

**Keywords:** serotonin, atherosclerosis, inflammation, 5-HT receptors

## Abstract

Serotonin, also known as 5-hydroxytryptamine (5-HT) is a well-known neurotransmitter in the central nervous system (CNS), but also plays a significant role in peripheral tissues. There is a growing body of evidence suggesting that serotonin influences immune cell responses and contributes to the development of pathological injury in cardiovascular diseases, such as atherosclerosis, as well as other diseases which occur as a result of immune hyperactivity. In particular, high levels of serotonin are able to activate a multitude of 5-HT receptors found on the surface of immune cells, thereby influencing the process of atherosclerotic plaque formation in arteries. In this review, we will discuss the differences between serotonin production in the CNS and the periphery, and will give a brief outline of the function of serotonin in the periphery. In this context, we will particularly focus on the effects of serotonin on immune cells related to atherosclerosis and identify caveats that are important for future research.

## 1. Introduction

Serotonin, also known as 5-hydroxytryptamine (5-HT), has been well-studied regarding the role it plays in both neurotransmission and mood disorders as part of the central nervous system (CNS). While serotonin is best known as a featured member of CNS transmission, it was initially discovered in the blood, as a regulator of vascular tone. The etymology of the name ‘serotonin’ is derived from ‘sero’ and ‘tonin’, which relate to the muscular tone in blood vessels, thus implicating it also in vasoreactivity, in addition to its role in neurotransmission [1,2,3]. In the periphery of the body, serotonin serves as a regulatory molecule, as well as a hormone, and plays a role in the functioning of the cardiovascular system due to its vasoreactive effects [4,5]. Additionally, there are differentially expressed receptors for serotonin present on immune cells, implicating it in immune functions, and together with its cardiovascular role, it is strongly implicated as a player in atherosclerosis. Overall, serotonin is a molecule that has versatile functions and can be implicated in many physiological and pathological functions. Although the role of serotonin as a regulator in the CNS has been extensively studied and is still debated, its role in the periphery remains even more unclear. Despite the crucial role it plays in neurotransmission in the CNS, 90% of serotonin found in the body is stored in the periphery [6].

In this review, we will focus on the functions of serotonin in the periphery of the body, with emphasis on its role, and the role of its receptors in vascular reactivity and immune cell reactivity—functions of serotonin that relate to but are not well-studied in the context of atherosclerosis. Thereafter, we will discuss current knowledge related to pharmaceutical serotonin regulation with relevance to cardiovascular disease risk and atherosclerosis.

## 2. Physiological Production of Serotonin: Serotonin Formation and Regulation

The role or serotonin as a neurotransmitter in the regulation of mood is well-established and has been well-characterized since its discovery. Our knowledge of the role of serotonin in the rest of the body is proving to have many complex threads, which are still being untangled for a better understanding of the precise mechanisms by which serotonin works. Serotonin is produced from the essential amino acid tryptophan, which can be metabolized in the body to produce bioactive compounds via three common routes: (i) the kynurenine pathway, which uses most of the bioavailable tryptophan, (ii) the indole pathway, as part of the immune system, which uses residual levels of bioavailable tryptophan, and (iii) the serotonin-producing pathway, which uses less than 10% of the bioavailable tryptophan in the body [6,7,8]. Serotonin in the CNS and periphery do not originate from the same source, as serotonin is a highly charged molecule that is unable to cross the blood–brain barrier [9].

The CNS produces its own serotonin for use as a neurotransmitter, as part of the neuronal system that regulates aspects of mood and behavior (see Figure 1). In the CNS, tryptophan is first hydroxylated by the enzyme tryptophan hydroxylase 2 (TPH2) [10], after which it is decarboxylated by the aromatic amino acid decarboxylase enzyme to form 5-HT/serotonin, before being packaged in vesicles for release [11,12,13]. Once serotonin has been produced and released, it is taken up for storage through serotonin transporter (SERT) channels, which are membrane transporter channels that allow the uptake of serotonin in serotoninergic neurons [14].

The formation of serotonin in the periphery of the body occurs by the same mechanism as in the CNS, but by means of the tryptophan hydroxylase 1 (TPH1) enzyme (see Figure 1). The bulk of serotonin found in the body is produced by means of the TPH1 enzyme in enterochromaffin cells, which are found in the mucosal lining of the gastrointestinal tract [15,16,17]. Once released from these cells into the bloodstream, serotonin is taken up for storage through SERT channels found on blood platelets for transport and delivered to the periphery of the body (see Figure 2). Interestingly, the TPH2 iso-enzyme is also locally present in the periphery of the body in the enteric nerves, which are found along the lining of the gut and regulate peristalsis and gastrointestinal tract motility. The enteric nerves thereby serve as an overlap point between the CNS and periphery for the production of serotonin.

The CNS and enterochromaffin cells play important roles in the production of serotonin. However, other cells/tissues are also able to produce serotonin; such as pancreatic B-cells and adipocytes, among others [18,19]. The presence of TPH1 in locations, such as B-cells and adipocytes, in addition to the stomach, supports the notion that serotonin plays a role in metabolism, as serotonin signaling would be able to originate and localize in these regions.

Production of serotonin in the body, as well as regulation of serotonin levels, are influenced by many factors, including genetics, diet, hormone levels, drug use, and function of the 5-HT transporters present in effector tissues [1]. For example, a study involving a weight loss intervention in obese children showed that plasma serotonin levels measured after weight loss were significantly reduced following a dietary lifestyle intervention that involved caloric restriction [20]. This illustrates the role that diet can play in the regulation of serotonin levels. Other factors, such as age and hormone levels, may also contribute, as indicated in the study by Gonzales et al., who measured serum estradiol levels in aged women and showed a comparative reduction in serotonin with these factors when compared to a control group [21]. Genetic variation is another factor that can contribute to serum serotonin levels, as illustrated in a study of individuals with psycho-social problematic traits, where some genetic variants of genes involved in the serotonin production system were shown to correlate with serum levels of serotonin [14].

Under basal conditions, serotonin is kept stored in cells and granules (see Figure 2), and plasma levels are relatively low (approx. 10 nM) [12]. When released into the peripheral blood, and by neurons in the CNS, the signaling activity of serotonin is regulated by the 5-HT receptors, which are predominantly G-protein coupled receptor types that vary with receptor isoform and by cell type. When activated, these receptors conduct signal cascades that promote various effects which will be discussed later [22,23]. The signaling process is terminated once serotonin is taken up into cells by the SERT receptor [24]. When this occurs, serotonin can either be stored and reused, or degraded into secondary products, such as melatonin. Serotonin is converted to the bioactive compound, melatonin, by means of a rate-limiting catalytic enzyme, serotonin N-acetyltransferase [25]. Melatonin, in turn, is also noted for its beneficial effects across a broad array of sleep and related disorders. As a highly charged molecule [9], the reuptake of serotonin into cells must occur in order for it to be modified or degraded. Once inside the cell, serotonin can be degraded by the monoamine oxidase enzyme [26]. In blood platelets, the uptake process of serotonin occurs in an autocrine manner, as part of a feedback loop (see review [27]); this occurs in the CNS too, although the regulation between the two locations is different in terms of response to bring SERT to the cell surface.

The release of serotonin from storage in platelets triggers a diverse array of effects which can include blood vessel dilatation or constriction, smooth muscle convulsion, cellular hypertrophy, or hyperplasia [28,29]. Signal transmission to elicit these effects relies on the 5-HT group of receptors. This group of receptors comprised seven classes of the 5-HT receptors, 5-HT1 through 5-HT7, each with their own subtypes, usually indicated by a letter (e.g., receptor subtype 5-HT1A, see Figure 3). These subtypes, while part of the same class and responsive to similar chemical agents, have distinctive features from each other and may be found in specific locations in the body, on specific cell types, or linked to particular G-protein coupled receptors (GPCRs) that facilitate their unique effects [23,30,31,32].

## 3. Serotonin and Vascular Reactivity

### 3.1. Serotonin Receptors

Most of the 5-HT receptors have been characterized as GPCRs, which initiate various intracellular signaling cascades, depending on the receptor class and the G-protein that couples to the receptor [33,34,35]. Each class of receptors is composed of receptors that have similar properties with regard to pharmacology, signal transduction, and structure. The 5-HT1 and 5-HT5 classes of receptors couple with Gi/o G-proteins, which inhibit adenylyl cyclase activity [36,37], while the 5-HT2 class of receptors preferentially couple with Gq/11 G-proteins, which are linked to inositol phosphate signaling and movement of cytosolic Calcium ions [38]. The 5-HT3 class of receptors is the only class of 5-HT receptors that are not classified as GPCRs, as they have been characterized as ligand-gated ion channel receptors which contain Cys-loop domains [10]. The 5-HT4, 5-HT6, and 5-HT7 receptor classes all couple preferentially to Gs G-proteins, which elicit signal transmission by means of cyclic adenosine monophosphate (cAMP) and protein kinase A (PKA) [39,40,41,42,43]. Additionally, the 5-HT4 receptor class is also noted to couple with the Gα13 protein and the 5-HT7 receptor class is noted to couple with the G12 protein, which both are known to play a role in long-term structural changes in cells [44]. Each of these subtypes can be associated with different cell types and elicit variable effects, based on location, vessel size, and health state of an individual, among other factors. Serotonin molecules present in the blood stream and neuronal junctions bind to these receptors and elicit a wide variety of downstream cellular responses related, for example, to neural conduction and vasoreactivity. Notably, serotonin receptors are also used as a target for the treatment of vasospasms in the pathogenesis of migraines, highlighting their role in vasoreactivity, even in the cerebral hemisphere (see review [23]).

The role of serotonin as a neurotransmitter has been extensively studied and reviewed (see reviews [45,46,47,48]). Therefore, this review will especially focus on the vascular and immunological effects of serotonin, rather than those involved in the CNS, to illustrate the role they play in vascular reactivity and immune cell responses, and thus their potential role as an agent in atherosclerosis.

### 3.2. The Development and Progression of Atherosclerosis

Atherosclerosis is a chronic inflammatory disease, characterized by a progressive buildup of lipids in the arterial walls [49]. The initial stages in the formation of an atherosclerotic plaque are due to shear stress. In vessels, this causes damage to the endothelial layer and becomes a site where lipids accumulate. Accumulation of lipids triggers inflammation in the endothelial layer, to which monocytes respond, after which they penetrate the intima and differentiate into macrophages [50]. These macrophages further develop into foam cells by the uptake and retention of cholesterol-containing low-density lipoproteins [51]. Over time, the continued aggregation of foam cells causes the migration of smooth muscle cells into the intima to provide structural support, and forms a fibrous cap around the fatty streak of foam cells along the lumen of the vessel [51]. This is known as a stable plaque, but as the lipid core grows, the fibrous cap encroaches the luminal space, narrows the arteries, and may become a calcified plaque that inhibits vasoreactivity [52]. These plaques can remain stable or, if the lipid core continues to grow, can become labile in response to stresses that weaken the fibrous cap, causing it to rupture more easily [53,54]. This, in turn, causes aggregation of blood platelets and thrombus formation which blocks arterial flow, leading to ischemia. In the clinical setting, this is the process that underlies both myocardial infarction and strokes as ischemia, due to atherosclerosis in coronary and cerebral arteries, can cut off blood flow to parts of the heart and brain, respectively, resulting in oxygen deprivation in these regions.

### 3.3. Early Studies Involving Serotonin and Vasoreactivity

Vasoreactivity constitutes the constriction or dilatation of blood vessels in response to stimuli. During vasoconstriction, the muscles in the vessel walls contract, causing the narrowing of the lumen, restricting blood flow, and increasing blood pressure. This can happen in response to local factors such as an increase in partial oxygen pressure, a decrease in temperature, and high concentrations of adrenalin or noradrenalin [55]. Vasoconstriction can also occur systemically in response to signaling by means of angiotensin II, vasopressin, endothelin A, thromboxane A2, prostacyclin, reactive oxygen species, and serotonin [55]. During vasorelaxation, the muscles in the vessel walls relax, causing the widening of the lumen, increasing blood flow, and decreasing blood pressure. Vasorelaxation can occur locally in response to decreased partial oxygen pressure, increased concentration of nitric oxide, an increase in temperature, an increase in adenosine, and response to low concentrations of adrenalin [55]. Additionally, vasorelaxation can also occur systemically in response to signaling by means of prostacyclin, endothelin B, bradykinin, kallikrein, histamine, and serotonin [55]. Vasoreactivity is a prominent feature in the progression of atherosclerosis, as atherosclerotic plaques affect vessel pliability, which can exacerbate ischemic events [56]. In the context of atherosclerosis, vasoreactivity is impaired because of calcification of the vessel walls as part of the development of an atherosclerotic plaque. Such calcification and development of the plaque cause vasoconstriction that cannot be reversed [56]. Given the dual role that serotonin is capable of playing in vascular reactivity, distinguishing between dilatation and constriction [56] serves as a logical starting point before delving into deeper, more complex functions of this molecule.

As part of the vascular system, reactivity to serotonin appears to be related to a number of factors, including the type of vessel (arterial or venous), the diameter of the vessel, the location of the vessel in the body (in essential organs or limbs), pathological insult, and the assortment of serotonin/5-HT receptors in the region [1]. For example, in smaller arterioles, serotonin elicits dilatation, while the effects in venules vary [1]. In the context of the vascular system, there are several studies, in both human and animal models, which have been used as exploratory tools to try and understand the effects that are seen in response to serotonin receptor agonists and antagonists, and thus to characterize the receptor subtypes and their associated functions.

In studies conducted in human vessels, manipulation of the 5-HT receptors by means of activators and inhibitors resulted in different responses, demonstrating functional differences between the receptors. One study used an ex vivo model to examine contractile responses in superficial human hand veins with known activation of 5-HT receptor classes 5-HT1–5-HT3. The outcomes of this study, which was performed in healthy individuals showing no symptoms of cardiovascular, metabolic, or connective tissue diseases, was that 5-HT1 and 5-HT2 receptor classes were involved in this contractile response, in a dose-dependent manner, but 5-HT3 receptors did not play a role in contractile responses in these vessels [57]. A second study, conducted in vivo, examined the vascular response to an administered combination of serotonin and a 5-HT3 receptor antagonist in the forearm veins of healthy individuals. Vasodilation occurred as a result, implicating 5-HT3 receptor activation as an inhibitor of dilatation [58].

Vessel-type and the state of health also seem to play a role in 5-HT receptor activity. In vitro examination of 5-HT receptor activation in human epicardial coronary artery segments, obtained from patients with ischemic heart disease and healthy controls, showed differential involvement of the 5-HT receptor classes in mediating vasoreactivity. In healthy tissue, 5-HT1 and 5-HT2 receptor classes mediated vasoconstriction in non-ischemic tissue, while ischemic tissue contraction was only in response to 5-HT1 activation, but not to 5-HT2 activation [59]. Interestingly, it could additionally be observed that vessel segments taken from ischemic tissue that were subsequently exposed to 5-HT1 receptor stimulation showed greater reactivity in the segments further from the occlusion, possibly implicating the 5-HT1 receptor group as a potential target in combating ischemia.

One of the earliest animal studies on the vasoreactive effects of serotonin examined the effect of intraperitoneal vs. intravenous administration of serotonin in rats. Intraperitoneal delivery of serotonin promoted a reduction in blood pressure, with no change in cardiac output (i.e., vasorelaxation) observed in the myocardium, pulmonary parenchyma, skeletal muscle, bone, and CNS [60]. There was, however, a decrease observed in blood flow to the kidneys and skin [60]. Intravenous administration increased cardiac output, blood pressure, and peripheral blood flow, suggesting a vasoconstrictive response by this route of administration. Responses noted in this study were observed in a dose-dependent manner.

More recent animal studies have been able to look more closely at the 5-HT receptor effects on vasoreactivity when activated in the CNS versus peripherally. One such study examined the effects of CNS vs. peripheral administration of a 5-HT2A/C receptor activator ((±)-1-(2,5-dimethoxy-4-iodophenyl)-2-aminopropane hydrochloride), capable of crossing the blood–brain barrier, on arterial pressure in conscious rats [61]. Intracerebroventricular and intravenous administration routes both produced elevated iliac arterial pressure (i.e., constriction) in response to the above-mentioned receptor agonist, but the delivery of a 5-HT2 receptor antagonist, xylamidine tosylate, that does not cross the blood–brain barrier prevented this response only with intravenous administration, but not with intracerebroventricular administration. This suggests that the two methods of administration elicit the same response possibly via different mechanisms in the brain and periphery. All in all, this study illustrated that 5-HT2 receptor activation plays a role in the elevation of iliac arterial pressure.

A follow-up study which also examined the serotonin receptors in vascular reactivity investigated the effect of serotonin administration on vascular response in conscious rabbits, with a combined administration of kentaserin and methiothepin (5-HT2 receptor antagonists) or methysergide (a 5-HT1 and 5-HT2 agonist) [62]. The observed contraction of the renal artery associated with 5-HT2 receptor activation and dilatation in renal, mesenteric, and hindquarter vascular beds is suggested to be mediated by 5-HT1 activation.

In summary, these early vascular reactivity studies illustrate the dynamic and variable effects that can occur with the activation of different 5-HT receptor classes (see Table 1). They also indicate that the same receptor class can be activated to produce different effects, based on vessel size and location, and the health state of the individual/affected tissue. These receptor classes can be broken down into subclasses for further examination, some of which appear to have more neuronal involvement, and others of which have both neuronal and peripheral effects. These receptor activities and interactions with serotonin and vascular walls appear to be location-specific and need to be investigated in the context of both location and pathology in order to better understand their functions and interactions.

## 4. Serotonin and Immune Response in Chronic Inflammation and Atherosclerosis

In the periphery of the body, once triggered by an injury, serotonin is mainly released from platelets into the circulatory system where it can elicit downstream effects. In the instance of acute or chronic inflammation, serotonin attracts neutrophils to sites of injury, which trigger an innate immune response [13]. Once released, serotonin interacts with 5-HT receptors which are present at the cell surfaces of tissues, as well as circulating cells that are part of the immune system. Several immune cells have been noted to express 5-HT receptors, with a large variation of receptor classes and sub-types depending on the cell type (recently elaborately reviewed by [63]). The interaction of serotonin with immune cells has dynamic and complex outcomes.

Circulating blood factors are noted to express 5-HT receptors, with various receptor subtypes associated with specific immune response cells. These interact with serotonin to elicit a multitude of responses. Below, a general overview is provided of these receptor subtypes and noted responses in immune cells.

### 4.1. B-Cells

B-cells, also known as B-lymphocytes, function as part of the adaptive immune system, and are responsible for antibody production and immune memory [64]. Atherosclerosis is a chronic inflammatory condition, and B-cells play a role in atherosclerotic plaque formation by the production of antibodies and cytokines within vessel walls [65]. Serotonin was shown to be able to increase mitogen-stimulated B-cell proliferation in a time and dose-dependent manner—a phenomenon that could be replicated with the administration of a 5-HT1A receptor agonist in splenocytes stimulated with lipopolysaccharide (LPS) and dextran sulfate to become mature B-lymphocytes [66]. Serotonin has been shown to upregulate B-cell proliferation by means of the 5-HT1A receptor (see Figure 4 for a summary of receptors with immune cell type) [67,68]. Receptor antagonist for 5-HT1A was able to block this proliferation, thus indicating that the effects of serotonin on B-lymphocytes are facilitated through the 5-HT1A receptors [66]. At the level of transcription, stimulation of splenocytes with LPS to form B-cells showed increases in 5-HT1A mRNA levels—a transcription event that could be blocked by inhibition of NF-κB [67]. This indicates that transcription of the 5-HT1A receptor in B-lymphocytes is likely facilitated through NF-κB transcriptional activity, linking it to mitogen-activated pathways.

A number of B-lymphocyte cell lines express SERT channel proteins and are able to take up serotonin from their environments [69]. Direct uptake of serotonin through the SERT channels also appears to have effects independent of the 5-HT receptors, as serotonin has been shown to induce apoptosis in Burkitt’s lymphoma—a phenomenon which was not inhibited by 5-HT receptor antagonists, but which was abolished by administration of selective serotonin reuptake inhibitors (SSRIs), which block the SERT channel [69,70]. However, this effect of serotonin does not affect healthy cell types, only cells associated with Burkitt’s lymphoma, further supporting the notion that serotonin has differential effects depending on the health state of the tissue with which it interacts. Furthermore, long-term treatment with SSRIs leads to an increase in B-lymphocytes in human patients, indicating that the regulation of serotonin goes together with immune regulation [71].

As part of the adaptive immune system, B-cells present antigens for the recruitment of other immune cells to neutralize non-self and damaged tissue. Of particular note is the interaction between B2-cells which present antigens for the recruitment of T-cells, which can play a role in vascular tissue invasion and the start of plaque formation [64]. The presence of serotonin plays a role in the maturation of B-cells implicating serotonin in adaptive immune reactivity. Furthermore, increased detection of SERT channels in pathologically affected tissue implicates the uptake of serotonin in the propensity of disease, although it is still unclear how this exactly occurs.

### 4.2. T-Cells

Another main cell type in adaptive immunity is T-cells, also known as T-lymphocytes, which are partially responsible for the recognition of antigens with immune memory. T-cells play a role in immune responses to peptide epitopes related to atherosclerosis [64], such as peptides derived from apolipoprotein B. They exist in the bone marrow in the naïve state and can under the influence of various stimuli differentiate to become T-helper cells, which activate macrophages and aid antibody responses, or cytotoxic T-cells, which target cells infected with intracellular microbes. T-cells are noted to predominantly express the 5-HT7 and low levels of the 5-HT1B receptors in the naïve state [72,73]. Stimulation of naïve T-cells with serotonin causes activation of T-cells, while antagonism of the 5-HT3 receptor inhibited activation of T-cells [74], implicating both serotonin and 5-HT3 as key players in T-cell activation. By contrast, a study in both human and murine primary T-helper cells by Yin et al. implicated 5-HT1B as an inhibiting agent in the proliferation of primary T-helper cells and T-cell lines, which indicates there may be a differential response in mature T-cells, depending on the cell type [75].

Serotonin has been shown to upregulate T-cell proliferation, by means of the 5-HT1A [67,68]. Both mRNA transcription, as well as cell surface presence and activity of 5-HT1A, were shown to be increased in splenocytes in mice when treated with mitogens, such as LPS, phytohemagglutinin, concavalin A, and phorbol 12-myristate 13-acetate with ionomycin, which stimulates T-cell proliferation [67]. Transcription of the 5-HT1A receptor was blocked by inhibition of NF-κB, implicating NF-κB as an important driver of 5-HT1A receptor expression in T-cells. These findings indicate that serotonin plays a role in the proliferation of T-cells as part of the immune system.

In addition to the combination of 5-HT receptors expressed by T-cells, they have also been shown to express the TPH1 enzyme and to produce serotonin [72]. There have been conflicting reports regarding the expression of SERT channel receptors for the uptake of serotonin on T-cells. León-Ponte Matilde et al. reported that SERT receptors were detected in differentiated regulatory T-cells, but not in effector T-cells, despite their common origin, while Wu et al. showed no detectable levels of SERT in naïve T-cells, nor in activated T-cells [72,73,74,75,76].

T-cells express 5-HT receptors and show reactivity to serotonin, as it causes them to mature, implicating serotonin as an important player in T-cell activation and as a candidate involved in the aberrant activity of T-cells, as seen in the formation of fatty deposits along vascular walls. Furthermore, T-cells also express the TPH1 enzyme, which would allow these cells to influence the activity of cells located in the nearby vicinity by the production and release of serotonin, to activate inflammatory signaling cascades.

### 4.3. Natural Killer Cells

Natural killer (NK) cells are part of the lymphocyte family and have an innate killing capacity. They are an element of the innate immune system that responds to infection and recognizes foreign antigens, as presented by viruses and bacteria [77]. NK cells act in response to antigen-presenting T-cells, as well as antigenic exogenous lipids, such as bacterial LPS, which is known to drive atherosclerotic plaque formation [78,79]. Once activated, NK cells release cytokines which further activate immune cell adhesion to promote atherosclerotic lesions (see review [80]). Serotonin interacts with monocytes and NK cells in such a way as to suppress the inflammatory activity of monocytes and allow NK cells to take on cytotoxic functions that would, under normal circumstances, be modulated by monocytes. Treatment of a combination of NK cells and monocytes with serotonin was shown to enhance NK cell cytotoxicity in response to antigen stimuli, in a dose-dependent manner [81]. This activity was reproduced when treating the cells with a 5-HT1A receptor agonist and was blocked by the treatment with a broad-spectrum 5-HT antagonist, demonstrating that the 5-HT1A receptor mediates this cytotoxic capability of NK cells. This effect, as elicited by NK cells, was dependent on the presence of monocytes and the release of a soluble factor into the cellular environment which was not clearly identified. These responses, such as cytotoxicity (as illustrated above), proliferation, and production of IFN-γ, require cell-to-cell contact with monocytes and NK cells and are mediated by means of the 5-HT1A receptors on NK cells [82,83].

The use of SSRI agents to block the uptake of serotonin via SERT channels in NK cells showed an enhancement of circulating NK cells with long-term treatment in patients (396 ± 101 cells/mL vs. 159 ± 30 cells/mL in healthy controls), indicating that serotonin is able to influence the proliferation of NK cells [71]. In vitro experiments with SSRI treatment in NK cells in samples taken from patients with long-term viral infection also support the activity of serotonin as a proliferative agent for NK cells, which would interact with viral antigens [84].

### 4.4. Platelets

Platelets are an integral carrier of serotonin in the periphery of the body, as they take up serotonin released from enterochromaffin cells lining the gut into the bloodstream. They serve as storage capacity which release serotonin in response to inflammation when they are stimulated, which allows for specific targeting of locations within vessels in the body, as well as to effector cells as part of the immune system [85]. Platelets play a role in atherosclerosis, as they drive thrombus formation, and also facilitate the recruitment of inflammatory immune cells toward lesion sites [86]. The mobility of platelets and their ability to target tissues make them a key player in pathological conditions, particularly those of a vascular nature. They have been implicated as a role player in coronary atherogenesis, and the release of serotonin by platelets is capable of stimulating thrombogenesis, mitogenesis, and proliferation of smooth muscle cells, all of which contribute to the development of atherosclerotic plaques [87].

During hemostasis—a process that keeps blood contained within damaged vessels, such as occurs with the rupture of atherosclerotic plaques resulting in vessel occlusion and ischemia—serotonin acts as an agent that enhances platelet activation and adhesion to form a clot. It does so by ‘serotonylation’ of GTPases, which is a process in which serotonin covalently binds to GTPases and renders them constitutively active [88,89]. This is a process that was also shown to be inactive in mice with *TPH1* knockout, which could not produce serotonin in the periphery of the body [88]. These mice bled four times longer than their wild type counterparts—a phenomenon that was reversed when they were given serotonin treatment in advance [88]. These *TPH1* knockout mice were less prone to the formation of thromboembolisms, implicating serotonin as an important player in thrombus formation in vessels after atherosclerotic plaque rupture [88].

The 5-HT2A receptor has been implicated as the key serotonin receptor on platelets which facilitate platelet aggregation in response to serotonin; however, other 5-HT receptors are also known to be expressed on platelets [88,90]. The 5-HT3A receptor, which is the only ligand-gated cationic channel receptor of all 5-HT receptors, has also been detected at the cell surface of platelets in blood samples taken from healthy volunteers, and is likely involved in the depolarization of platelets (as it is thought to do in the CNS), though its function in platelets has not been clearly defined [90].

Platelets are known to play a role in other inflammatory diseases, such as rheumatoid arthritis [91,92]. In the setting of rheumatoid arthritis, platelets have been detected, in some instances in the vicinity of the junctions between endothelial cells in the synovial vessel lumen [93]. They have also been shown to adhere to circulating neutrophils and monocytes and to promote the adhesion of these cells to the bone socket endothelium [13,94]. These interactions are parallel to those of the development of atherosclerosis and could be a means by which plaque formation on vessel walls is formed and encouraged. Serotonin has also been implicated in these inflammatory and adhesion processes, as levels of serotonin detected in platelets derived from blood samples were markedly lower in patients suffering from rheumatoid arthritis when compared to those in remission and healthy individuals [92,95]. A study conducted in SERT-deficient mice showed that endothelial gaps in the synovial cavity were abrogated when blood platelets were absent [96]. Furthermore, this effect could be mimicked by the administration of SERT receptor blockers, thus implicating serotonin in pathology with these gap formations in rheumatoid arthritis [96]. In healthy individuals, serotonin is released in response to injury and triggers an inflammatory cytokine cascade meant to initiate repair of tissue by pathways involving physiological fibrosis—a mechanism which, when aberrant, could potentially underlie some instances of plaque formation.

### 4.5. Eosinophils

Eosinophils, a type of white blood cell, are components of the innate immune system that play a role in the phagocytosis of foreign bodies and the release of enzymes involved in allergic and parasitic immune reactions. Eosinophils are linked to the activity of platelets in the setting of atherosclerosis, as the two cell types mutually activate each other, which ultimately leads to plaque formation and thrombosis [97].

For example, interactions with platelets induce the formation of eosinophil extracellular traps which are present in thrombi. Furthermore, inhibition of 5-HT2A receptors with cyproheptadine could prevent the recruitment of eosinophils under shear stress, implicating theses receptors in the recruitment and function of eosinophils [98]. In animal studies involving allergic asthmatic reaction, transfusion of platelets from wild type mice to mice that were genetically modified to be *TPH-1* deficient were able to produce an allergic reaction with the transfer of platelets that contained serotonin. Stimulation of the 5-HT2A was also able to induce the migration of eosinophils, implicating them in eosinophil activity [99]. A study of allergy-induced asthma in mice showed that inhibition of both 5-HT1 and 5-HT2 by means of chemical antagonists was able to partially rescue the hyper-responsive airway reaction in these mice [100].

### 4.6. Neutrophils

Neutrophils are part of the innate immune system and play a role in the process of phagocytosis in the immune response. Neutrophils are recruited by interactions with platelets that release C-C motif chemokine ligand 5 (CCL5) which promote the adhesion of immune cells to vessel walls [101]. Neutrophils secrete granule proteins that promote macrophage polarization toward a pro-inflammatory phenotype and the cascade of immune cell events involved in plaque formation [102].

Release of serotonin from platelets induces the recruitment of neutrophils in the setting of inflammation, although so far, exact details concerning this mechanism and the receptors involved in this process remain elusive [13]. Serotonin released from platelets attracts neutrophils, possibly as a targeting mechanism in acute inflammation, and induces neutrophil degranulation and the release of radical oxygen species [103]. In the context of ischemia/reperfusion, particularly with myocardial infarction, the release of serotonin from neutrophils leads to enhanced inflammation and increased cell death [103]. Furthermore, patients with acute coronary syndrome showed elevated levels of serotonin, correlating with plasma levels of myeloperoxidase, which is released by neutrophils [103]. In line with this, long-term SSRI treatment in patients and animal models showed lower levels of myeloperoxidase and reduced neutrophil degranulation in the pathophysiological setting of the acute coronary syndrome [103]. This illustrates that there is some sort of interaction between serotonin and neutrophils; however, it is still unknown as to whether this is a direct interaction and which receptors are involved in this interaction.

### 4.7. Mast Cells

Mast cells, also known as mastocytes or labrocytes, are another component of the innate immune system noted to release heparin, histamine, and serotonin as part of immune response processes. They also produce leukotrienes, prostaglandins, and platelet-activating factors. Mast cells are implicated in atherosclerotic plaque formation, as they have been shown to enhance lipid uptake and can be engulfed by macrophages to form foam cells [104,105]. Mast cells can also affect the stability of atherosclerotic plaques by the recruitment of other inflammatory cells [106].

Mast cells reside predominantly in connective tissues of the body. Activity and degranulation of mast cells are known to regulate aspects such as vasodilation, vascular homeostasis, immune response, and angiogenesis [107,108,109,110]. They have also been implicated in the pathophysiology of allergic responses, asthma, anaphylaxis, gastrointestinal disorders, and cardiovascular diseases, among other conditions [111,112,113,114,115,116,117].

Mast cells are able to both synthesize and store serotonin [118]. Serotonin in itself is able to induce adherence of mast cells to fibronectin and can induce migration of mast cells by means of chemotaxis to serotonin for the promotion of inflammation, as observed in the context of dermal tissue injury in mice [119]. After screening of an array of 5-HT mRNA transcripts in mast cells, mRNA transcripts were detected for 5-HT1A, 1B, 1E, 2A, 2B, 2C, 3, 4, and 7 [119]. The 5-HT1A receptor was identified as the key serotonin receptor which elicits responses in mast cells, as administration of the receptor agonist for the 5-HT1A was able to nullify the adhesion and migration responses described above. This finding was consistent in vivo, as *5-HT1A* knockout mice that were injected with serotonin showed no migratory or adhesive responses with regard to mast cells [119]. Thereby, serotonin serves as a means to attract mast cells to sites of inflammation.

A recent genetic study that obliterated the synthesis of serotonin in mice by means of genetic deletion of the TPH1 enzyme specifically in mast cells showed that these mice were protected from developing obesity and insulin resistance, both of which are strongly associated with the exacerbation of atherosclerosis. Taken in context, this study highlights that cell type specificity of serotonin production may affect disease pathology related to immunity [120].

Mast cells express many 5-HT receptors and may have multiple effects in response to serotonin stimulation, likely based on environmental factors. In contrast to other cell types, serotonin is noted to play an anti-atherogenic role in the context of mast cells, preventing adhesive events in vascular walls, and thus helping to prevent atherosclerosis. More studies are needed to fully elucidate the effects of serotonin (and its receptors) on mast cells in contrast to other immune cell types.

### 4.8. Monocytes

Monocytes are the largest in size of the leukocytes and serve as precursors which may differentiate to become macrophages or dendritic cells. During atherogenesis, monocytes infiltrate the vessel wall and differentiate to become macrophages, which engulf oxidized lipoproteins to form lipid-rich foam cells. In this way, monocytes contribute to the development of atherosclerotic plaques.

Analysis of mRNA expression in LPS-stimulated human blood monocytes revealed expression of 5-HT1E, 2A, 3, 4, and 7 [121]. Activation of the 5-HT4 and 7 receptor subtypes by means of serotonin triggered intracellular cyclic AMP (cAMP) signaling, which stimulated mRNA expression and subsequent cellular release of interleukin (IL)-1β, IL-6, IL-8/CXCL8, IL-12p40, and tumor necrosis factor (TNF)-α, which favor T-helper 2 cells that aid B-cell antibody secretion, as would be necessary for the development of atherosclerosis [121], while it had no effect on the production of IL-18 and interferon (IFN)-γ, which favor T-helper 1 cells involved in delayed-type hypersensitivity reactions [121]. Further investigation into the mRNA transcripts of these released factors showed that LPS stimulation linked to the 5-HT3 receptor, by means of differential responses to agonists and antagonists of the 5-HT3 receptor, specifically induced production of IL-1β, IL-6, and IL-8/CXCL8, but had no notable effect on the production of TNF-α and IL-12p40 with LPS stimulation [121]. Activation of the 5-HT4 and -7 by means of receptor agonists increased release of IL-1β, IL-6, IL-12p40, and IL-8/CXCL8, but had no notable effect on the release of TNF-α with LPS stimulation [121]. This study also showed, with the use of agonists and antagonists, that the effects of serotonin to induce cytokine release from monocytes happened by means of 5-HT3, -4, and -7, while 5-HT1E and -2A did not modulate cytokine production or release in these monocytes.

### 4.9. Macrophages

Macrophages are a component of the innate immune system, most prominently noted for the role they play in phagocytosis [122]. Secondary to this, they are, for example, also involved in aspects of antigen presentation, cytokine secretion, the release of granulocyte and monocyte colony-stimulating factors, the release of interferons, and cytokines, which include TNF-α, TNF-β, and transcription growth factor (TGF) β, thus triggering inflammatory events and the neutralization of foreign bodies [123]. In the context of atherogenesis, classically activated M1 macrophages play a central role in the development of plaques, as they promote and sustain inflammation [124].

Macrophage phagocytosis in bone morrow is known to be modulated by INF-γ. The presence of serotonin together with IFN-γ has shown differential outcomes with regard to phagocytosis based on the concentration of serotonin in the region—higher physiological concentrations impaired phagocytosis, while lower doses promoted it [125]. Serotonin in itself has been shown to increase the phagocytotic activity of macrophages in vitro—an effect that was replicated with the administration of a 5-HT1A receptor agonist and abolished with the administration of a 5-HT1A receptor antagonist [126]. An interesting finding in this set of experiments was that an inhibitor of NF-κB was also able to replicate the results of the 5-HT1A receptor agonist, linking this transcription factor to 5-HT1A activity [126]. IFN-γ, when co-administered with serotonin, is able to induce phagocytosis in murine bone marrow-derived macrophages, but when treated with antagonists of 5-HT1A, -2A, and -2C receptors, this phagocytotic response was nullified [125]. These findings implicate these receptors in IFN-γ-induced phagocytosis in macrophages and highlight the idea that serotonin is likely to work in concert with other signaling molecules to induce specific responses in each unique context [125]. These data provide functional evidence that at least part of the modulation of IFN-γ-induced phagocytosis by 5-HT occurs through a 5-HT receptor-mediated mechanism, and 5-HT, dopamine, and histamine modulate IFN-γ-induced phagocytosis independently through their respective receptors.

There appears to be a differential expression of serotonin receptors on macrophages, based on the location and maturation state of the macrophages in question. When examining the role of serotonin in macrophage polarization from human monocytes in vitro, administration of serotonin in the presence of LPS caused an upregulation of genes associated with M2 anti-inflammatory macrophages (*SERPINB2*, *THBS1*, *STAB1*, and *COL23A1*) and decreased the expression of genes associated with M1 pro-inflammatory macrophages (*INHBA*, *CCR2*, *MMP12*, *SERPINE1*, *CD1B*, and *ALDH1A2*) [127]. LPS is a commonly used agent for the differentiation and activation of macrophages from preceding cell types. Serotonin was able to prevent the initiation of an inflammatory response in this context by impeding the release of cytokines IL12p40 and TNF-α [127]. This M2 polarization with the administration of serotonin was shown to be facilitated by 5-HT2B and -7 receptors, as detected by mRNA expression, whereas only the 5-HT7 receptor was shown to mediate an inhibitory action on the release of pro-inflammatory cytokines in M1 pro-inflammatory monocyte-derived macrophages [127].

Alveolar macrophages, which are the primary immune cells present in the pulmonary vessels, show a propensity to a different expression pattern of 5-HT receptors compared to those expressed on the monocyte-derived macrophages discussed above. When isolated and cultured, treatment of these macrophages with serotonin induces a rise in intracellular Ca^2+^—an event that is blocked by treatment with a 5-HT2C receptor inhibitor [128]. This observation was supported by in vivo experiments in 5-HT2C knockout mice which, when administered serotonin, did not show transcription responses to produce the CCL2 protein in alveolar macrophages, as had occurred in wild type counterparts [128]. CCL2 is a small inducible cytokine, regulated by circadian rhythm, which serves to attract monocytes, T-cells, and dendritic cells during inflammatory or infectious responses [129]. Cardiovascular events which are related to plaque rupture and vascular blockade are noted to have circadian susceptibility, with more serious incidences occurring in the morning hours. Blockade of CCL2 signaling has been shown to abolish the adhesion of leucocytes to the endothelial wall, a finding that can help to inhibit atherosclerosis [129]. Combined, these findings indicate that serotonin is able to influence inflammatory/immune responses in macrophages and can thereby influence atherosclerosis development.

### 4.10. Dendritic Cells (DCs)

DCs are antigen-presenting cells that are able to trigger the adaptive immune response for inflammation to occur. DCs accumulate in atherosclerotic lesions and engage in diverse pathogenic and protective mechanisms during atherogenesis. DCs contribute to early foam cell formation, regulate lipid metabolism, and control pro- and anti-atherosclerotic T-cell responses by various mechanisms (see review [130]).

The effects of serotonin on DCs are still poorly understood, although serotonin has already been shown to alter the differentiation of monocytes into DCs (as opposed to macrophages) [131]. This differentiation event was blocked by treatment with serotonin after incubation with a 5-HT1/-6/-7 multi-receptor antagonist and was able to replicate the effects of serotonin with the administration of a 5-HT1/-6/-7 multi-receptor agonist [131]. Furthermore, these DCs derived by serotonin stimulation show enhanced capacity for the release of cytokines when compared to a control group that was not treated with serotonin [131].

There appears to be a considerable variation in the expression of 5-HT receptors, based on the cell type from which the DCs are derived, and the inflammatory markers present in the environment. For instance, monocyte-derived DCs distinctly express higher levels of 5-HT2B receptor RNA in the CD1a-DC subset, which does not produce the pro-inflammatory cytokine IL-12p70, when compared to the CD1a+ subset which produces IL-12p70 [132]. Thus, higher levels of 5-HT2B, as expressed by DCs, may well be linked to anti-inflammatory activity. Additionally, there is a distinction between immature and mature DCs with regard to the 5-HT receptor types that are expressed on the cell surface. Immature cells are known to preferentially transcribe mRNA for the expression of 5-HT1B, 1E, 2A, and 2B receptors, which are linked to the movement of calcium within the cell. On the other hand, mature DCs mainly transcribe mRNA for 5-HT4 and 7, which are linked to cAMP elevation, as well as the release of cytokines IL-1β and IL-8, while reducing the secretion of IL-12 and TNF-α [133]. No mRNA transcripts for 5-HT1A, 1D, 1F, 2C, 5, and 6 receptors were found in DCs, and there were no noted changes in transcription levels of 5-HT2A and 3 with the maturation of DCs [133].

Stimulation of mature human DCs with serotonin, or with agonists of the 5-HT4 and 7, showed an in vitro increase in levels of cAMP, which is thought to play a supportive role in the recruitment of monocytes to vessel walls and their differentiation into macrophages in atherogenesis [133]. Treatment of mature cells with serotonin, and with 5-HT4 and 7 agonists, induced the release of pro-inflammatory cytokines IL-8 and IL-1β which serve to attract and activate neutrophils in inflammatory regions, an event that contributes to vascular plaque formation [133]. These pro-inflammatory responses were not observed with serotonin treatment in immature DCs, indicating that there is a differential expression of 5-HT receptors between the immature and mature cell states [133]. Further investigation with agonists for the 5-HT1B receptor and 5-HT1E/F receptors showed comparable results, implicating that the 5-HT1B and 1E/F receptors are also involved in the pathway causing a release of pro-inflammatory cytokines [133]. Treatment with agonists for 5HT3, 7, 4, and -1 also showed increased release of IL8 in mature DCs, while 5-HT1B, 1E/F, and 2 receptor agonists did not produce an increased response in IL-8 and IL-1β levels. It is noted that 5-HT4 and 7 receptors are coupled to G-proteins that produce cAMP, thus implicating them in pro-atherogenic activity [133]. Furthermore, treatment of mature DCs with increasing concentrations of serotonin showed an inverse relationship with detectable levels of TNF-α and IL-12 by these cells, and as higher levels of these two cytokines are noted to impair LPS-induced inflammation [134], this would mean that low levels of serotonin would be associated with less inflammation and less immune cell migration and infiltration in localized regions.

DCs, much like B-cells, are also able to take up serotonin directly from the environment through SERT receptors, although this process only occurs in mature DCs [135]. Much like B-cells, the capacity of DCs to take up serotonin was inhibited by the addition of SSRI agents, thus indicating that the SERT channel is functional and present on the surface of DCs. This expression of the SERT channel appears to be regulated by cell maturation state (with detectable expression levels of SERT only in mature DCs), interaction with T-cells (which are known to produce serotonin directly), and response to microbial stimuli, though further research is needed to verify and clarify these mechanisms [135]. Based on the above threads of information, taken together in the context of atherosclerosis, it seems that serotonin activity when coupled with DCs serves as an exacerbating factor in the development of inflammation underlying atherosclerosis.

## 5. Current Knowledge about the Impact of Serotonin Modifying Drugs and Cardiovascular Events

Given the flexibility of action of serotonin, it seems reasonable that drugs that modify serotonin would have multi-faceted effects as part of the regulation of the CNS, vascular tone, and immune responses. It would follow that modification of serotonin and its associated signaling could have an impact on cardiovascular health, directly, through inflammation and blood circulation, and indirectly, through alleviation of psychiatric distress, which is known to have an impact on health and physical wellbeing.

Selective serotonin reuptake inhibitors (SSRIs) are commonly used as a treatment for depression, which is known to be a risk factor for cardiovascular diseases (CVDs) [136]. There have been conflicting reports on the outcomes of SSRI treatment in cardiovascular pathologies, mainly because of the effects on atherosclerotic disease development.

In animal studies, one study looked at *Apoe*-deficient mice, a strain known to develop atherosclerosis when administered a high-fat diet, with long-term SSRI treatment. This study showed a progressive increase in atherosclerotic lesion formation by the enhancement of leukocyte adhesion to arterial walls [137]. Studies conducted in non-human primates with the treatment of depression with the SSRI, setraline, found that treatment of depression with SSRIs resulted in an almost 5-fold increase in coronary artery atherosclerosis when compared to untreated depressed monkeys, and a 6-fold increase in SSRI-treated depressed monkeys when compared to untreated wild type controls [138,139]. These results would indicate that (1) depression promoted the development of atherosclerotic plaque formation, (2) treatment with the SSRI agent further amplified this in depressed animals, and (3) this effect appeared to be independent of other traditional cardiovascular risk factors [138]. By contrast, a multi-ethnic long-term trial conducted in patients on a variety of anti-depressants, which included SSRIs, found their results did not support the long-term anti-depressant usage and exacerbation of sub-clinical atherosclerosis [140], highlighting the contrast in reports between beneficial and detrimental outcomes with modification of serotonin in the clinical setting.

Clinical studies which provide information on the mechanisms related to CVDs are likely to involve depression, which essentially simulates high circulating levels of serotonin, while treatment with SSRIs in relation to CVDs seems to be complex. SSRI treatment has been investigated in various scenarios related to inflammatory diseases, including CVDs, as they have been noted to have anti-inflammatory properties as part of depression treatment [141,142,143]. Some studies have reported a reduced risk of myocardial infarction with SSRI agents, and an increased risk with other non-SSRI anti-depressant agents [144].

A retrospective clinical study that evaluated the outcomes of adverse events with the use of SSRI agents reported an association between long-term antidepressant usage and elevated risks of coronary heart disease, CVD mortality, and all-cause mortality [145]. This finding was consistent with a second study which found an associated increase in stroke risk and all-cause mortality with SSRI usage [146], but no association between SSRI usage and coronary heart disease [146].

Additionally, meta-analyses have been conducted on studies evaluating the relationship between depressed patients treated with SSRI agents and major cardiovascular events (MACE). One such analysis looked at the association between major adverse cardiovascular events and the use of SSRIs in patients with previous cardiovascular events—their findings indicated that depressed patients treated with SSRIs have significantly reduced risk of myocardial infarction and MACE [147]. However, there is still much debate over the effectivity of SSRIs in reducing the risk of MACE overall, as both beneficial and adverse cardiovascular events can occur following the chronic use of various types of SSRIs [147], and this must be accounted for when considering patient treatment programs. A second analysis of studies from an array of databases (MEDLINE (PubMed), EMBASE, PsycINFO, CENTRAL, and the Cochrane Controlled Clinical Trial Register), in which patients with coronary heart disease and depression were treated with SSRI medication, indicated an overall decrease in depression and decrease in CVD-related re-admissions and mortality rates [148]. SSRIs have been associated with a significantly lower risk of myocardial infarction in patients with coronary artery disease and depression, and also in patient’s post-acute coronary syndrome with depression [149]. However, much like the contrast seen in animal studies, there has also been meta-analyses that have reported that SSRIs were associated with significant increases in all-cause mortality and the risk of cardiovascular events [150]. They also reported that the use of SSRIs (as well as other anti-depressant agents) were less harmful in patients who had pre-existing cardiovascular illnesses, and they posited that this may be due to the anti-clotting properties of these drugs.

A systematic review of the use of SSRI agents with CVD noted that some types of SSRIs, as reported in clinical intervention studies, prevented platelet adhesion and aggregation (i.e., citalopram and setraline) [151,152], and better controlled the cardiovascular risk profile, including insulin resistance (i.e., sibutramine), and body weight (i.e., sibutramine), while also inhibiting inflammatory processes (i.e., sibutramine and setraline) [153]. However, they also reported that cardiac arrythmia, tachycardia, and long quite time (QT) syndrome [154] were among the cardiac-related adverse reactions with the use of SSRIs (i.e., fluoxetine, sertraline, and citalopram), though these effects were reported in a relatively low number of patients [153].

There is much conflicting information on the outcomes of SSRI treatment in depressed patients with cardiovascular illness and adverse events. It would be prudent to consider the effects of SSRIs in concert with co-medications and the physiological state of the body with regard to immunity and inflammation. This could be interpreted as the idea that the effects of serotonin, as seen in the body, are not due to serotonin alone, but rather a combination of factors that would need to be teased apart to better explain how, and why, these conflicting findings occurred. The interplay between inflammatory disease and high circulating levels of serotonin/depression may well be circular—depression is noted to follow severe inflammatory diseases, such as inflammatory bowel disease, diabetes, and arthritis [155,156,157]. Inflammation plays a role in the development of depression, as evidenced by clinical and laboratory findings which demonstrate that a combination of SSRI anti-depressants and ant-inflammatory treatments accelerates the antidepressant activity [158,159].

## 6. Conclusions and Perspectives on Serotonin in Inflammatory Diseases

One factor to consider when evaluating the effects of serotonin is the expression of 5-HT receptors at the cell surface. Due to the consistent presence of 5-HT receptors on the cell surfaces of immune response cells, it is evident that serotonin can play a prominent role in immune activity. It is also directly implicated in auto-immune pathologies, though these would need to be individually evaluated to determine whether serotonin plays a pathological role and is a causative agent. In favor of this dual-nature hypothesis, depending on the physiological state, is the observation that serotonin modulates circulating immune response factors and is implicated in regulating the release of cytokines such as INF-γ, TNF-α, IL6, and IL-1β [121,160]. Given the vast and adaptive nature of responses in which serotonin is involved in the circulatory immune system, it seems sensible that it works in concert with a variety of other signaling molecules in order to amplify their responses, or to elicit a specific cooperative response when working in concert with particular cofactors. Perhaps there is cofactor binding at surface GPCRs, the combination of which would determine which 5-HT receptors are bound, and which effector pathway is activated.

A second factor to consider, that relates strongly to the above, would be the health state of a body when examining the effects of serotonin. Serotonin, like many other innate signaling molecules, is vital for physiological function and healing, but when dysregulated in response to pathophysiological signals, results in detrimental outcomes, such as atherosclerosis. Logic would dictate that these detrimental outcomes are facilitated by the 5-HT receptors expressed by cells, although the details of this dysregulation are yet to be resolved.

An additional factor to bear in mind when considering the effects of serotonin in pathophysiology is quantity. Under normal, basal conditions, serotonin is stored within cells, but when it is released in the blood, it is able to trigger signaling cascades and cellular effects, both locally and globally. Serotonin, in normal homeostatic conditions, is not meant to be present at high concentrations in the blood for extended periods of time. A constant higher level of serotonin detectable in the bloodstream can occur for a number of reasons, most commonly SERT receptor malfunction. In the clinical setting, this basal high level of serotonin in the bloodstream has been associated with a variety of inflammatory diseases including obesity, asthma, Crohn’s disease, and diabetes. In patients with rheumatoid arthritis, an inflammatory disease that causes erosion of the synovial joints, levels of serotonin in plasma were found to be elevated with the progression of joint erosion [161]. Similarly, patients with allergic asthma have elevated plasma levels of serotonin, and treatment with an agent that enhances serotonin uptake was able to reduce the severity of the asthmatic symptoms, thus indicating that serotonin plays a causal role in this auto-immune/inflammatory condition [162,163]. Irregularly high serum levels of serotonin have also been noted with obesity, where obese patients showed double the blood level of serotonin when compared to healthy controls [164,165]. Higher levels of both serotonin and tryptophan also corresponded with the development of diabetic complications in diabetic patients when compared to healthy controls [164,165]. Interestingly, in the instance of inflammatory bowel diseases, there is a clear distinction in circulating levels of serotonin between patients with Crohn’s disease and those with ulcerative colitis, as patients with Crohn’s disease have higher circulating levels of serotonin, which was not the same for patients with ulcerative colitis [166].

Serotonin appears to serve as an activation agent for processes of inflammation under normal physiological conditions, and as an intensifying pro-inflammatory agent in pathophysiological conditions that involve inflammation, such as atherosclerosis.

## Figures and Tables

**Figure 1 ijms-24-01549-f001:**
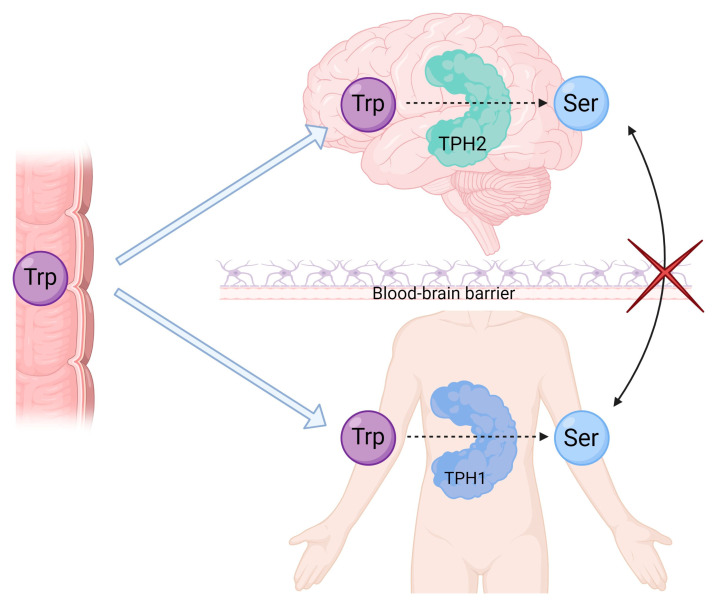
Formation of serotonin in the central nervous system (CNS) and periphery of the body. Tryptophan is absorbed through the intestines and is transported to the CNS, as it is non-polar and is able to cross the blood–brain barrier. Serotonin, which is a hydrophilic molecule and is thus unable to cross the blood–brain barrier, is produced separately in the CNS and the periphery through the process initiated by the tryptophan hydroxylase (TPH) enzyme specific to either region—TPH1 in the periphery of the body and TPH2 is specifically in the CNS. Created with BioRender.com.

**Figure 2 ijms-24-01549-f002:**
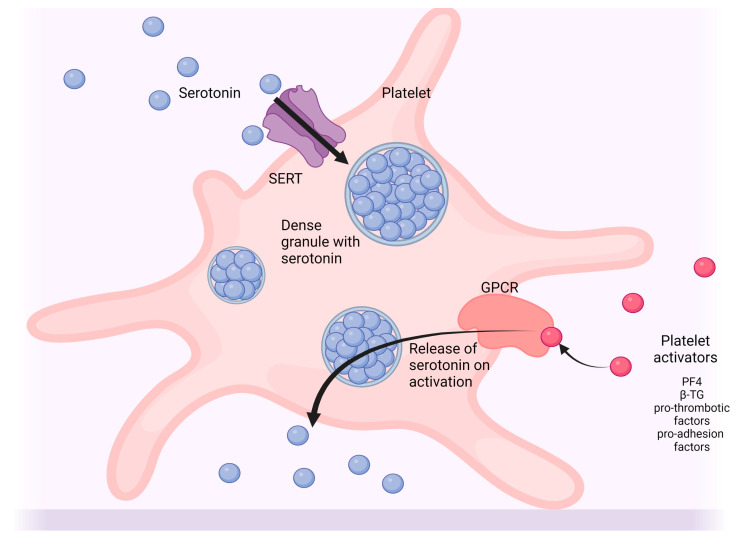
Uptake and release of serotonin by blood platelets. Serotonin is taken up from the bloodstream by the serotonin transporter (SERT) channels where they are stored in dense granules in blood platelets. When platelets are activated by platelet activators, such as platelet factor 4 (PF4), pro-thrombotic factors, pro-adhesion proteins, or β-thromboglobulin (β-TG) (among others), binding of the platelet activators to G-protein coupled receptors (GPCR) at the cell surface trigger a cascade of events which result in the release of serotonin from platelets into the periphery. Created with BioRender.com.

**Figure 3 ijms-24-01549-f003:**
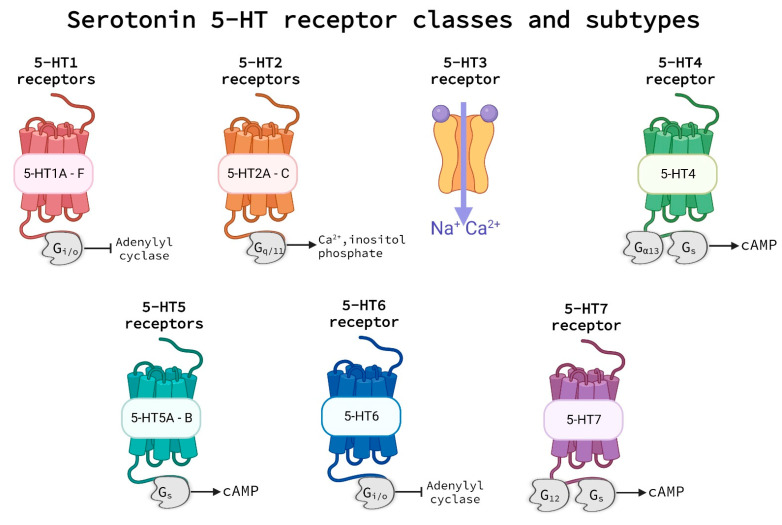
5-HT receptor groups and subtypes. All 5-HT receptors are G-protein coupled receptors with the exception of the 5-HT3 receptor, which is a ligand-gated channel. Serotonin binds to the 5-HT receptors outside of the cell triggering effector events via various G-protein-triggered signaling pathways. Abbreviations: 5-HT—5-hydroxytryptamine and cAMP—cyclic adenosine monophosphate. Created with BioRender.com.

**Figure 4 ijms-24-01549-f004:**
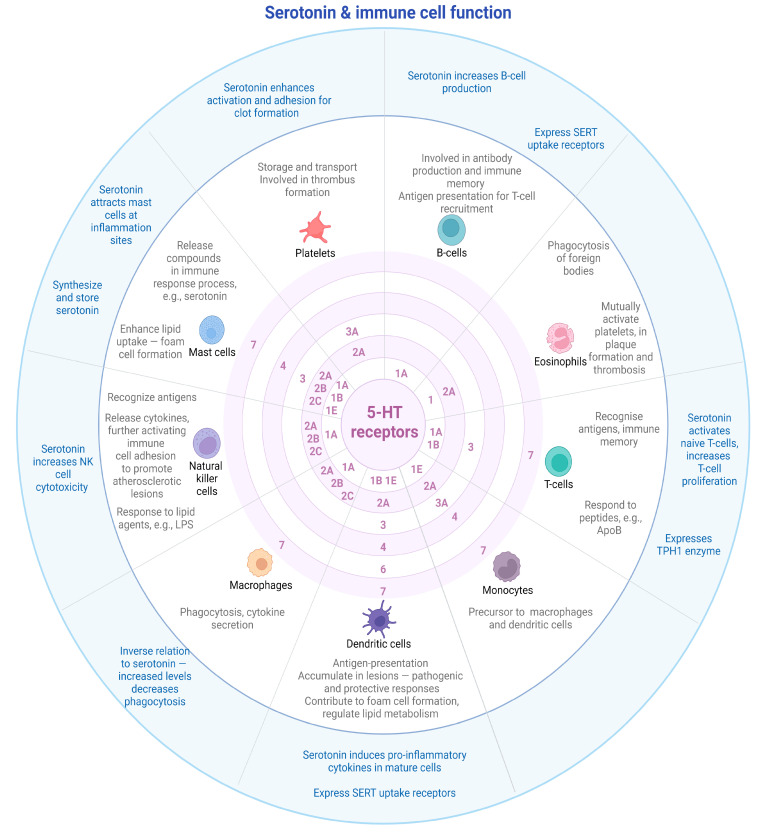
Expression of 5-HT receptor groups and subtypes on immune cells, and effects of serotonin on cellular function. Original references for the receptors as summarized in this figure are detailed in the main body of the text. Abbreviations: 5-HT—5-hydroxytryptamine, LPS—lipopolysaccharide, NK—natural killer, SERT—serotonin transporter, and TPH—tryptophan hydroxylase. Created with BioRender.com.

**Table 1 ijms-24-01549-t001:** Vasoreactivity to serotonin.

Receptor	Vasoreactivity	Location	Vessel Type	Reference
5-HT1	Constriction	Human superficial hand vein segments ex vivo	Venous	[57]
	Contraction in both healthy and ischemic vessels	Epicardial coronary artery in vitro	Arterial	[59]
	Dilatation	Renal, mesenteric, and hindquarter vessels in rabbits	Unspecified	[62]
5-HT2	Constriction	human superficial hand vein segments ex vivo	Venous	[57]
	Contraction in healthy but not ischemic vessels	Epicardial coronary artery in vitro	Arterial	[59]
	Constriction	Renal artery in rabbits	Arterial	[62]
	Constriction	Iliac artery	Arterial	[61]
5-HT3	Dilatation	Healthy individuals—venous flow in forearm vessels	Venous	[58]
	No significant effect	human superficial hand vein segments ex vivo	Venous	[57]
	No significant effect	Epicardial coronary artery in vitro	Arterial	[59]

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
