# Peer review of "Exploring the Role of Serotonin as an Immune Modulatory Component in Cardiovascular Diseases"

_ijms, 2023, doi:10.3390/ijms24021549_

Round 1

Reviewer 1 Report

This review is an overview of 5-HT receptors in the immune cells and also considers some cardiovascular and endothelial actions. In general it is an interesting review, but it could benefit from greater consideration of the large degree of work about serotonin receptors in the gastrointestinal tract and enteric nervous system where these interact with immune cells. 

Major points

1. Tryptophan is an amino acid and thus a polar molecule (Fig 1 legend) and probably more polar than serotonin which is hydrophilic rather than highly charged as distinct from phrases in introduction/figure legends and also elsewhere in text. These basic errors in describing the chemistry of these molecules need to be corrected throughout.

2. Consider adding the G protein(s) relaying 5-HT receptor signals in Fig 3 to make it more comprehensive and to assist authors in describing signal pathways below that will assist readers in their understanding of 5-HT receptor pharmacology. 

3. Further consideration should be given to the enormous amount of pharmacological studies undertaken in the last 25 years or so that are concisely summarized in IUPHAR receptor and ion channel pages so the 5-HT receptors and ligands are correctly described

4. line 158 refers to one out of many recent reviews describing 5-HT receptors as neurotransmitters - consider adding a range of these reviews to allow readers easy access to a broader range of study than covered here.

5. The different 5-HT receptors use different G proteins to relay their signals - correct section around line 148 to reflect this. Consult IUPHAR pages for guidance.

6. The section following lines 248-250 is mainly a summary of the immune cell types containing 5-HT receptors. This section would benefit from integration as at the moment it reads like a list and does not bring additional understanding to the topic. One way would be to have a diagram showing how activation of the different receptors is regulated on the separate immune cells (perhaps showing immune cell actions) replacing the tabulated list of receptors in Fig 4 (this fig adds little to the text). Perhaps the authors could think about how increases in 5-HT following an infection may modulate the immune or wound responses ....

7. Concluding section - note that inflammatory bowel disease are different from irritable bowel syndrome. line 652 - irritable bowel syndrome is associated with depression but is not directly considered an inflammatory disease. line 695 - replace irritable with inflammatory - irritable bowel syndrome is a different condition without marked inflammation associate with Chron's or ulcerative colitis

minor points

line 137 - geted should read 'gated'.

line 635 Apoe shoould read ApoE

line 653 - ant-inflammation should read anti-inflammation

Reviewer 2 Report

IJMS Serotonin 2022

This review article looks at the role of peripheral serotonin acting on immune cells and affecting the development and progression of atherosclerosis and hence cardiovascular disease events.

The paper is well written and has some nice figures. There some issues with balance of the title and the content in that there is very little on actual atherosclerosis in the paper just lots of speculation some stated and some inferred.

For balance and context (for the reader) the paper needs a short succinct section on the current hypotheses around the development and progression of atherosclerosis – from the initial inflammatory response of cytokine and growth factor modification of the lipid binding glycosaminoglycan chains on proteoglycans to lipid binding to modified proteoglycans, through the later inflammatory stages of monocyte infiltration and macrophage development, other immune cell involvement and foam cell formation, along with smooth muscle cell migration and proliferation, collagen synthesis and finally the development of stable and labile plaques.

The paper also needs a specific section on the current knowledge about the impact of serotonin modifying drugs and cardiovascular events in patients  – there are many such drugs and there must be data available on their impact directly and indirectly (via alleviating depression) on CVD events.

Mentioning role of serotonin in vasospasm and migraine would be useful.

Specific

Line 46:pharmaceutical should perhaps be pharmacological

Throughout – the role of serotonin in CNS conditions is not as clear and well established as stated in this paper – in some areas it is hotly contested - please soften a little to give correct context.

119: “serum levels are low” – I believe that plasma levels are low but serum is a clotted blood product and when blood clots the serotonin is released from platelets so serum levels can be very high.

163: section on vasoreactivity and atherosclerosis  – this is unclear (and there are no refences) – see comments above on cell biology of atherosclerosis - there is a massive area about endothelial dysfunction (and hence vascular reactivity) and also the role of nitric oxide although the link directly between reactivity and atherosclerosis is otherwise somewhat obscure – please provide more insights in this area..

165: “binary starting point” – what does this mean? – please expand and explain.

END
